# TDR-CL: Targeted Doubly Robust Collaborative Learning for Debiased Recommendations

**Haoxuan Li**[1]   **Yan Lyu**[1]   **Chunyuan Zheng**[2]   **Peng Wu**[3*]

[1]Peking University
[2]University of California, San Diego
[3]Beijing Technology and Business University
{hxli, lyuyan}@stu.pku.edu.cn, czheng@ucsd.edu, pengwu@btbu.edu.cn

## Abstract

Bias is a common problem inherent in recommender systems, which is entangled with users' preferences and poses a great challenge to unbiased learning. For debiasing tasks, the doubly robust (DR) method and its variants show superior performance due to the double robustness property, that is, DR is unbiased when either imputed errors or learned propensities are accurate. However, our theoretical analysis reveals that DR usually has a large variance. Meanwhile, DR would suffer unexpectedly large bias and poor generalization caused by inaccurate imputed errors and learned propensities, which usually occur in practice. In this paper, we propose a principled approach that can effectively reduce the bias and variance *simultaneously* for existing DR approaches when the error imputation model is misspecified. In addition, we further propose a novel semi-parametric collaborative learning approach that decomposes imputed errors into parametric and nonparametric parts and updates them collaboratively, resulting in more accurate predictions. Both theoretical analysis and experiments demonstrate the superiority of the proposed methods compared with existing debiasing methods.

## 1 Introduction

Addressing various tasks in recommender systems (RSs) with causality-based methods has become increasingly popular (Wu et al., 2022b). Causality-based recommendation has shown its great potential in both numeric experiments and theoretical analyses across extensive literature (Chen et al., 2020; Wang et al., 2019). Generally, the basic question faced in RS is that "what would the feedback be if recommending an item to a user", requiring to estimate the causal effect of a recommendation on user feedback. To answer the question, many methods have been proposed, such as inverse propensity score (IPS) (Schnabel et al., 2016), self-normalized inverse propensity score (SNIPS) (Swaminathan & Joachims, 2015), error imputation based (EIB) methods (Steck, 2010), and doubly robust (DR) methods (Chen et al., 2021; Wang et al., 2019; 2021; Dai et al., 2022; Ding et al., 2022). Among them, the DR method and its variants show superior performance. We compare and evaluate these methods in terms of three desired properties, including doubly robust (Hernán & Robins, 2020; Wu et al., 2022c), robust to small propensities (Rosenbaum, 2020), and low variance (Tan, 2007). Failing to meet any of them may lead to sub-optimal performance (Molenberghs et al., 2015; van der Laan & Rose, 2011). Our theoretical analysis shows that DR has much greater variance and is less robust to small propensities compared to EIB (Kang & Schafer, 2007), even though the imputed errors and the learned propensities are accurate. Meanwhile, DR would suffer unexpectedly large bias and poor generalization caused by inaccurate imputed errors and learned propensities, which usually occur in practice.

In this paper, we first propose a novel targeted doubly robust (TDR) method, that can capture the merits of both DR and EIB effectively, by leveraging the targeted learning technique (van der Laan & Rose, 2011; 2018). TDR can effectively reduce the bias and variance *simultaneously* for existing DR approaches when the imputed errors are less accurate. Remarkably, TDR provides a model-

---

*Corresponding author.

agnostic framework and can be assembled into any DR method by updating its error imputation model, resulting in more accurate predictions.

To further reduce the bias and variance during the training process, we propose a novel uniform-data-free TDR-based collaborative learning (TDR-CL) approach that decomposes imputed errors into a parametric imputation model part and a nonparametric error part, where the latter adaptively rectifies the residual bias of the former. By updating the two parts collaboratively, TDR-CL achieves a more accurate and robust prediction. Both theoretical analysis and experiments demonstrate the superiority of TDR and TDR-CL compared with existing methods.

## 2 PRELIMINARIES

Many debiasing tasks in RS can be formulated using the widely adopted potential outcome framework (Neyman, 1990; Rubin, 1974). Denote $\mathcal{U} = \{u\}$, $\mathcal{I} = \{i\}$ and $\mathcal{D} = \mathcal{U} \times \mathcal{I}$ as the sets of users, items and user-item pairs, respectively. Let $x_{u,i}$, $r_{u,i}$, and $o_{u,i}$ be the feature, feedback, and exposure status of user-item pair $(u, i)$, where $o_{u,i} = 1$ or $0$ represents whether the item $i$ is exposed to user $u$ or not. Define $r_{u,i}(1)$ as the potential outcome if $o_{u,i}$ had been set to 1, which is observed only when $o_{u,i} = 1$. In RS, we are often interested in answering the causal question: "if we recommend products to users, what would be the feedback?". This question can be formulated as to learn the quantity $\mathbb{E}(r_{u,i}(1)|x_{u,i})$, i.e., it requires to predict $r_{u,i}(1)$ using feature $x_{u,i}$, where $\mathbb{E}$ denotes the expectation with respect to the target distribution $\mathbb{P}$. Many classical tasks in RS can be defined as estimating this quantity, such as rating prediction (Schnabel et al., 2016) and post-click conversion rate prediction (Guo et al., 2021). More examples can be found in Wu et al. (2022b).

Let $f_\theta(x_{u,i})$ be a model used to predict $r_{u,i}(1)$ with parameter $\theta$. Ideally, if all $r_{u,i}(1)$ for $(u, i) \in \mathcal{D}$ were observed, $\theta$ can be trained directly by optimizing the following ideal loss

$$\mathcal{L}_{ideal} = |\mathcal{D}|^{-1} \sum_{(u,i)\in\mathcal{D}} e_{u,i},$$

where $e_{u,i}$ is the prediction error, e.g., the squared loss $e_{u,i} = (r_{u,i}(1) - f_\theta(x_{u,i}))^2$. However, since $r_{u,i}(1)$ is observed only when $o_{u,i} = 1$, the ideal loss is non-computable. Restricting the analysis to non-missing data will obtain biased conclusions, as the observed data may form an unrepresentative sample of the target population. Different debiasing methods are designed to approximate and substitute the ideal loss. For example, the IPS and EIB estimators are given as

$$\mathcal{L}_{IPS} = |\mathcal{D}|^{-1} \sum_{(u,i)\in\mathcal{D}} o_{u,i} e_{u,i}/\hat{p}_{u,i}, \quad \mathcal{L}_{EIB} = |\mathcal{D}|^{-1} \sum_{(u,i)\in\mathcal{D}} [o_{u,i} e_{u,i} + (1 - o_{u,i})\hat{e}_{u,i}],$$

where $\hat{p}_{u,i}$ is an estimate of propensity score $p_{u,i} := \mathbb{P}(o_{u,i} = 1|x_{u,i})$, $\hat{e}_{u,i}$ is an estimate of prediction error $g_{u,i} := \mathbb{E}[e_{u,i}|x_{u,i}]$, i.e., it fits $e_{u,i}$ using $x_{u,i}$. The DR estimator is formulated as

$$\mathcal{L}_{DR} = |\mathcal{D}|^{-1} \sum_{(u,i)\in\mathcal{D}} \left[ \hat{e}_{u,i} + \frac{o_{u,i}(e_{u,i} - \hat{e}_{u,i})}{\hat{p}_{u,i}} \right],$$

which enjoys doubly robust property, i.e., it is an unbiased estimator of ideal loss when either imputed errors or learned propensities are accurate.

## 3 MOTIVATION

DR approaches have been extensively studied in RS for various debiasing tasks for its double robustness, e.g., rating prediction (Wang et al., 2019; 2020a; Li et al., 2023b;c), learning-to-rank (LTR) (Saito, 2020; Oosterhuis, 2022), and post-click conversion rate prediction (Guo et al., 2021; Dai et al., 2022), etc. However, the DR still have several limitations that need to resolved. We first show that DR has a large variance and is sensitive to small propensities as shown in Proposition 1 (see Appendix A for proofs).

**Proposition 1.** *If $\hat{p}_{u,i}$ and $\hat{e}_{u,i}$ are accurate estimates of $p_{u,i}$ and $g_{u,i}$, respectively, i.e., $\hat{p}_{u,i} = p_{u,i}$, $\hat{e}_{u,i} = g_{u,i}$, then IPS, EIB and DR estimators are unbiased, and their variances satisfy*

$$\mathrm{Var}(\mathcal{L}_{EIB}) \leq \mathrm{Var}(\mathcal{L}_{DR}) \leq \mathrm{Var}(\mathcal{L}_{IPS}),$$

*where the equality holds if and only if $p_{u,i} = 1$ for all $(u, i) \in \mathcal{D}$. In addition, when $p_{u,i}$ tends to 0, $\mathrm{Var}(\mathcal{L}_{IPS})$ and $\mathrm{Var}(\mathcal{L}_{DR})$ tends to infinity, and $\mathrm{Var}(\mathcal{L}_{EIB})$ tends to its minimum.*

Proposition 1 shows that the EIB estimator is low-variance and robust to small propensities (Tan, 2007; Imbens & Rubin, 2015; Wu et al., 2021; 2022a). In RS, some small propensities will appear inevitably due to the sparsity of the exposed data, resulting in a significant difference between $\mathrm{Var}(\mathcal{L}_{EIB})$ and $\mathrm{Var}(\mathcal{L}_{DR})$. Nevertheless, EIB usually has a large bias and is not preferred in practice. Proposition 1 provides a motivation to develop an estimator that combines the low-variance and robustness to small propensities of EIB with the double robustness of DR.

In summary, DR outperforms IPS in terms of both bias and variance. When compared with EIB, if $\hat{e}_{u,i}$ is inaccurate but $\hat{p}_{u,i}$ is accurate, DR tends to have a smaller bias, but if both $\hat{e}_{u,i}$ and $\hat{p}_{u,i}$ are accurate, then EIB has a smaller variance. If $\hat{e}_{u,i}$ is accurate but $\hat{p}_{u,i}$ is inaccurate, then EIB may be superior to DR in terms of both bias and variance. In practice, both $\hat{p}_{u,i}$ and $\hat{e}_{u,i}$ are likely to be at least mildly inaccurate, so choosing from EIB and DR involves the bias-variance trade-off. Ideally, it is desirable to develop a method that is robust to small propensities, with lower bias and variance compared to previous DR methods, while maintaining the double robustness.

## 4 COLLABORATIVE LEARNING DEBIASING FRAMEWORK

### 4.1 TARGETED DOUBLY ROBUST ESTIMATOR

We first bridge the explicit form of the DR estimator and the EIB estimator by noting that

$$\mathcal{L}_{DR} = \underbrace{\frac{1}{|\mathcal{D}|} \sum_{(u,i)\in\mathcal{D}} [o_{u,i}e_{u,i} + (1 - o_{u,i})\hat{e}_{u,i}]}_{\mathcal{L}_{EIB}} + \underbrace{\frac{1}{|\mathcal{D}|} \sum_{(u,i)\in\mathcal{D}} o_{u,i}(e_{u,i} - \hat{e}_{u,i})\frac{1 - \hat{p}_{u,i}}{\hat{p}_{u,i}}}_{\text{correction term}}, \quad (1)$$

where $\mathcal{L}_{DR}$ is formally equivalent to adding a correction term using learned propensities to $\mathcal{L}_{EIB}$. The correction term has an important role in the bias-variance trade-off for the estimations of the ideal loss as shown in Proposition 1. Specifically, compared with $\mathcal{L}_{EIB}$, $\mathcal{L}_{DR}$ can reduce bias by adding the correction term. As a compromise, the correction term will increase the variance of the DR estimator. Thus, if $\hat{e}_{u,i}$ is computed in a manner that ensures that

$$\frac{1}{|\mathcal{D}|} \sum_{(u,i)\in\mathcal{D}} o_{u,i}(e_{u,i} - \hat{e}_{u,i})\frac{1 - \hat{p}_{u,i}}{\hat{p}_{u,i}} = 0. \quad (2)$$

then the EIB estimator would have small bias and the DR estimator would have small variance.

For equation (2) to hold, a naive method is taking it as a constraint condition when training the error imputation model. However, the constraint (2) may degrade the accuracy of the imputed errors because it will restrict the hypothesis space of the error imputation model. Instead of directly estimating $\hat{e}_{u,i}$ satisfying the constraint (2), we propose to exploit the extra information on propensities when training the error imputation model. The basic idea of the proposed TDR estimator consists of the following two steps.

**Step 1 (Initialization).** Let $\hat{e}_{u,i}$ be the imputed error obtained by using any of the existing DR methods.

**Step 2 (Targeting).** Update $\hat{e}_{u,i}$ by fitting an extended one-parameter model as follows

$$\tilde{e}_{u,i}(\eta) = \hat{e}_{u,i} + \eta(1/\hat{p}_{u,i} - 1) \quad (3)$$

which includes a single variable $1/\hat{p}_{u,i} - 1$ and the offset $\hat{h}(x_{u,i})$. The parameter $\eta$ is solved by minimizing the squared loss between $\tilde{e}_{u,i}(\eta)$ and $e_{u,i}$ in the exposed events. Then the proposed TDR estimator is given as

$$\mathcal{L}_{TDR} = |\mathcal{D}|^{-1} \sum_{(u,i)\in\mathcal{D}} \left[ \tilde{e}_{u,i} + o_{u,i}(e_{u,i} - \tilde{e}_{u,i})/\hat{p}_{u,i} \right].$$

The targeting step enlarges the hypothesis space of $\tilde{e}_{u,i}$ compared to $\hat{e}_{u,i}$, and does not sacrifice the accuracy of the error imputation model, due to the introduce of an error correction term $1/\hat{p}_{u,i} - 1$ to estimate $e_{u,i}$. Theorem 1 shows the validity and preservation of TDR (see Appendix B for proofs).

**Theorem 1.** *The imputed error $\tilde{e}_{u,i}$ obtained with TDR satisfies the following properties:*
*(a) (validity) $\tilde{e}_{u,i}$ satisfies equation (2), which implies TDR would have smaller bias than EIB and smaller variance than DR based on the initial imputed error $\hat{e}_{u,i}$.*
*(b) (preservation) $\hat{\eta}$ in the targeting step will converge to 0 and renders $\tilde{e}_{u,i} = \hat{e}_{u,i}$ when $\hat{e}_{u,i}$ already satisfies equation (2).*

From Theorem 1, TDR guarantees that equation (2) always holds, regardless of the choice of the initial imputed errors. In addition, TDR inherits the desirable properties of EIB, such as low-variance and robust to small propensities, since equations (1) and (2) implies that the TDR estimator can be regarded as an EIB estimator.

TDR would reduce the variance of DR as shown in Theorem 1, a further question is whether the variance-reduction will come at the expense of an increase in bias? Remarkably, TDR has no sacrifice of bias. Specifically, it can be shown (see Appendix C) that the bias of both $\mathcal{L}_{DR}$ and $\mathcal{L}_{TDR}$ are composed of the product of the errors of the propensity model and imputation model weighted by $1/\hat{p}_{u,i}$. Therefore, given the same learned propensities, the more accurate the imputed errors are, the smaller the bias is. Since TDR updates $\hat{e}_{u,i}$ by adding an extra term $1/\hat{p}_{u,i} - 1$, so $\tilde{e}_{u,i}$ is expected to be more accurate than $\hat{e}_{u,i}$, resulting in a smaller bias for $\mathcal{L}_{TDR}$ than $\mathcal{L}_{DR}$.

Importantly, the TDR provides a model-agnostic framework due to the free choice of the initial imputed errors in Step 1, which has great potentially strengths for recommendation. TDR can be assembled into any competing DR approach (Wang et al., 2019; Guo et al., 2021; Dai et al., 2022), by updating its error imputation model with the targeting step. This extra targeting step tends to reduce both the bias and variance of the competing DR approach, resulting in more accurate predictions. Next, Theorem 2 indicates the double robustness of the TDR estimator (see Appendix C for proofs).

**Theorem 2.** *The proposed TDR estimator have the following properties:*
*(a) (unbiasedness under accurate imputed errors) $\mathcal{L}_{TDR}$ is unbiased if $\tilde{e}_{u,i}$ accurately estimates $g_{u,i}$.*
*(b) (unbiasedness under accurate learned propensities) Suppose that $\hat{p}_{u,i}$ accurate estimates $p_{u,i}$, and the validity of $\hat{e}_{u,i}$ doesn't hold, then $\mathcal{L}_{EIB}$ is biased, while $\mathcal{L}_{TDR}$ is unbiased.*

Besides, Theorem 2(b) reveals that TDR can remove the bias of $\mathcal{L}_{EIB}$ even though the initial imputed errors are inaccurate, provided the learned propensities are accurate.

## 4.2 SEMI-PARAMETRIC COLLABORATIVE LEARNING

In this subsection, we propose a novel TDR-based collaborative learning (TDR-CL) approach, in which the imputed errors $\tilde{e}_{u,i}$ are decomposed into a parametric error imputation model part $\hat{e}_{u,i}$ and a nonparametric targeting part $\omega_{u,i} \triangleq \eta(1/\hat{p}_{u,i} - 1)$ as in Section 4.1, i.e., $\tilde{e}_{u,i} = \hat{e}_{u,i} + \omega_{u,i}$. The latter corrects the residuals of the error imputation model. By updating both the parametric and nonparametric parts collaboratively, the bias and variance of the TDR estimator can be further reduced, resulting in more accurate predictions.

First, the embedding of each user $u$ and item $i$ is obtained by matrix factorization, and the stack layer gets the embedding $x_{u,i}$ by concatenation. TDR-based learning methods require estimated propensities for all user-item pairs, thus the Naive Bayes approach is no longer applicable. To handle this problem, the pre-trained propensities are obtained by conducting logistic regression of $o_{u,i}$ on $x_{u,i}$, and the model parameters are used as the initialization of $p_\xi(x_{u,i})$ in the iterative learning process. Given both the parametric error imputation part $\hat{e}_{u,i} = g_\phi(x_{u,i})$ and nonparametric targeting part $\omega_{u,i}$, the propensity model $p_\xi(x_{u,i})$ and the prediction model $f_\theta(x_{u,i})$ are updated simultaneously using the training loss

$$\mathcal{L}_{TDR-CL}(\theta, \xi, \phi) = \mathcal{L}_{TDR} + |\mathcal{D}|^{-1} \sum_{(u,i) \in \mathcal{D}} \Big[ -o_{u,i} \cdot \log \hat{p}_{u,i} - (1 - o_{u,i}) \cdot \log(1 - \hat{p}_{u,i}) \Big],$$

where $\hat{p}_{u,i} = p_\xi(x_{u,i})$, $e_{u,i} = (f_\theta(x_{u,i}) - r_{u,i}(1))^2$, $\tilde{e}_{u,i} = (f_\theta(x_{u,i}) - g_\phi(x_{u,i}) - \omega_{u,i} - \perp(f_\theta(x_{u,i})))^2$ with $\perp$ the operator that sets the gradient of the operand to zero thus $\nabla_\theta \perp (f_\theta(x_{u,i})) = 0$ and $\perp(f_\theta(x_{u,i})) = f_\theta(x_{u,i})$.

Then, unlike traditional alternative learning algorithms that directly use the parametric part $g_\phi(x_{u,i})$ as $\tilde{e}_{u,i}$, the proposed collaborative learning additionally uses $\omega_{u,i}$ as a non-parametric correction

---

**Algorithm 1:** The Proposed Targeted Doubly Robust Collaborative Learning, TDR-CL

---

**Input:** observed ratings $\mathbf{R}^o$, pre-trained learned propensities $\hat{\mathbf{P}}$, and $\omega_{u,i} = 0$.

1 **while** *stopping criteria is not satisfied* **do**
2      **for** *number of steps for training the prediction and propensity model* **do**
3          Sample a batch of user-item pairs $\{(u_j, i_j)\}_{j=1}^J$ from $\mathcal{D}$;
4          Update $\theta$ and $\xi$ by descending along the gradient $\nabla_{\theta,\xi} \mathcal{L}_{TDR-CL}(\theta, \xi, \phi)$;
5      **end**
6      **for** *number of steps for training the imputation model with targeting step* **do**
7          Sample a batch of user-item pairs $\{(u_k, i_k)\}_{k=1}^K$ from $\mathcal{O}$;
8          Update $\phi$ by descending along the gradient $\nabla_\phi \mathcal{L}_e(\theta, \xi, \phi)$;
9          Sample a batch of user-item pairs $\{(u_l, i_l)\}_{l=1}^L$ from $\mathcal{D}$;
10         $\eta^* \leftarrow \arg\min_\eta \sum o_{u,i}(e_{u,i}(\theta) - \hat{e}_{u,i}(\phi) - \eta(1/\hat{p}_{u,i} - 1))^2$;
11         Update $\omega_{u,i} \leftarrow \omega_{u,i} + \eta^*(1/\hat{p}_{u,i} - 1)$ for all user-item pairs.
12      **end**
13 **end**

---

term summed with $g_\phi(x_{u,i})$ to correct the estimation of $e_{u,i}$. Specifically, given the prediction model and the propensity model, $\tilde{e}_{u,i}$ first updates its parametric part $g_\phi(x_{u,i})$ by minimizing

$$\mathcal{L}_e(\theta, \xi, \phi) = |\mathcal{D}|^{-1} \sum_{(u,i) \in \mathcal{D}} o_{u,i}(\tilde{e}_{u,i} - e_{u,i})^2 / \hat{p}_{u,i},$$

where $e_{u,i} = r_{u,i}(1) - f_\theta(x_{u,i})$, $\tilde{e}_{u,i} = g_\phi(x_{u,i}) + \omega_{u,i}$. Next, the targeting step described in Section 4.1 is applied to further update the imputed errors $\tilde{e}_{u,i}$. Through calculating the optimal step size for line search $\eta^* = \arg\min_\eta \sum o_{u,i}(e_{u,i}(\theta) - \hat{e}_{u,i}(\phi) - \eta(1/\hat{p}_{u,i} - 1))^2$, the non parametric targeted error term $\omega_{u,i}$ is updated by adding $\eta^*(1/\hat{p}_{u,i} - 1)$.

In summary, the proposed learning approach collaboratively update the parametric term $\hat{e}_{u,i} = g_\phi(x_{u,i})$ and the nonparametric term $\omega_{u,i}$ to achieve a better trade-off to estimate $e_{u,i}$, which can reduce the bias of the existing DR methods such as DR-JL (Wang et al., 2019) and MRDR-DL (Guo et al., 2021), by further modeling for the fitted residuals of the parametric parts $\hat{e}_{u,i}$. On the other hand, as shown in Theorems 1 and 2, when $\hat{e}_{u,i}$ is already an accurate estimate of $e_{u,i}$, the introduction of the targeted error term $\tilde{e}_{u,i}$ satisfies no-harm property and the unbiasedness is maintained. We summarized the proposed TDR-CL approach in Alg. 1.

## 5   SEMI-SYNTHETIC EXPERIMENTS

In this section, following the previous studies (Schnabel et al., 2016; Wang et al., 2019; Saito, 2020; Guo et al., 2021), we aim to answer the following research question (RQ) on the semi-synthetic datasets:

**RQ1.** Does the proposed TDR estimator in estimating the ideal loss have both the statistical properties of lower bias and variance in the presence of selection bias?

### 5.1   EXPERIMENTAL SETUP

**Dataset and Preprocessing. MovieLens 100K[1] (ML-100K)** is a dataset of 100,000 missing-not-at-random (MNAR) ratings from 943 users and 1,682 movies collected from movie recommendation ratings. **MovieLens 1M[2] (ML-1M)** is a larger dataset of 1,000,209 MNAR ratings from 6,040 users and 3,952 movies. Following the data preprocessing procedure of previous studies (Schnabel et al., 2016; Wang et al., 2019; Saito, 2020; Guo et al., 2021), we first use matrix factorization (Koren et al., 2009) to complete the rating matrix in the five-scale. Then for each predicted ratings $R_{u,i} \in$

---

[1]https://grouplens.org/datasets/movielens/100k/
[2]https://grouplens.org/datasets/movielens/1m/

Table 1: Mean and standard deviation of the relative error on the Naive, EIB, IPS, DR and TDR.

| Dataset | Method | ONE | THREE | FIVE | ROTATE | SKEW | CRS |
|---------|--------|-----|-------|------|--------|------|-----|
| ML-100K | Naive | $0.0688 \pm 0.0025$ | $0.0790 \pm 0.0028$ | $0.1027 \pm 0.0028$ | $0.1378 \pm 0.0011$ | $0.0265 \pm 0.0021$ | $0.1062 \pm 0.0022$ |
| | EIB | $0.5442 \pm 0.0016$ | $0.5878 \pm 0.0017$ | $0.6167 \pm 0.0018$ | $0.2533 \pm 0.0004$ | $0.3584 \pm 0.0007$ | $0.1443 \pm 0.0007$ |
| | IPS | $0.0338 \pm 0.0033$ | $0.0390 \pm 0.0037$ | $0.0511 \pm 0.0033$ | $0.0696 \pm 0.0026$ | $0.0129 \pm 0.0027$ | $0.0526 \pm 0.0026$ |
| | DR | $0.0140 \pm 0.0034$ | $0.0180 \pm 0.0037$ | $0.0150 \pm 0.0034$ | $0.0401 \pm 0.0016$ | $0.0101 \pm 0.0027$ | $0.0237 \pm 0.0025$ |
| | TDR | $\mathbf{0.0053 \pm 0.0026^{*}}$ | $\mathbf{0.0035 \pm 0.0025^{*}}$ | $\mathbf{0.0066 \pm 0.0032^{*}}$ | $\mathbf{0.0325 \pm 0.0015^{*}}$ | $\mathbf{0.0029 \pm 0.0020^{*}}$ | $\mathbf{0.0193 \pm 0.0025^{*}}$ |
| ML-1M | Naive | $0.0682 \pm 0.0007$ | $0.0783 \pm 0.0007$ | $0.1014 \pm 0.0008$ | $0.1377 \pm 0.0005$ | $0.0256 \pm 0.0007$ | $0.1054 \pm 0.0006$ |
| | EIB | $0.5437 \pm 0.0005$ | $0.5872 \pm 0.0005$ | $0.6157 \pm 0.0005$ | $0.2531 \pm 0.0001$ | $0.3575 \pm 0.0002$ | $0.1442 \pm 0.0001$ |
| | IPS | $0.0343 \pm 0.0009$ | $0.0394 \pm 0.0009$ | $0.0508 \pm 0.0009$ | $0.0687 \pm 0.0006$ | $0.0130 \pm 0.0008$ | $0.0528 \pm 0.0007$ |
| | DR | $0.0130 \pm 0.0009$ | $0.0168 \pm 0.0009$ | $0.0133 \pm 0.0009$ | $0.0399 \pm 0.0005$ | $0.0090 \pm 0.0008$ | $0.0229 \pm 0.0007$ |
| | TDR | $\mathbf{0.0054 \pm 0.0009^{*}}$ | $\mathbf{0.0031 \pm 0.0009^{*}}$ | $\mathbf{0.0076 \pm 0.0009^{*}}$ | $\mathbf{0.0324 \pm 0.0005^{*}}$ | $\mathbf{0.0031 \pm 0.0008^{*}}$ | $\mathbf{0.0187 \pm 0.0007^{*}}$ |

Note: * means statistically significant results (p-value $\leq 0.001$) using the paired-t-test compared with the best baseline.

$\{1, 2, 3, 4, 5\}$, we assign the $p_{u,i} \in (0, 1)$ with $p_{u,i} = p\alpha^{\max(1, 5 - R_{u,i})}$. Finally, we replace the predicted ratings $R_{u,i}$ with $r_{u,i}^{\text{true}}$ in $\{0.1, 0.3, 0.5, 0.7, 0.9\}$ and sample the binary click indicator and conversion label with the Bernoulli sampling $o_{u,i} \sim \text{Bern}(p_{u,i}), r_{u,i} \sim \text{Bern}(r_{u,i}^{true}), \forall (u, i) \in \mathcal{D}$, where $\text{Bern}(\cdot)$ denotes the Bernoulli distribution.

**Predicted Metrics.** The following prediction metrics are used to evaluate the debiasing performance under different scenarios.

• **ONE:** $\hat{r}_{u,i}$ is identical to the $r_{u,i}^{\text{true}}$, except that $|\{(u, i) \mid r_{u,i}^{\text{true}} = 0.9\}|$ randomly selected $r_{u,i}^{\text{true}}$ of $0.1$ are flipped to $0.9$.

• **THREE:** Same as ONE, but flipping $r_{u,i}^{\text{true}}$ of $0.3$ instead.

• **FIVE:** Same as ONE, but flipping $r_{u,i}^{\text{true}}$ of $0.5$ instead.

• **ROTATE:** $\hat{r}_{u,i} = r_{u,i} - 0.2$ when $r_{u,i} \geq 0.3$, and $\hat{r}_{u,i} = 0.9$ when $r_{u,i} = 0.1$.

• **SKEW:** $\hat{r}_{u,i}$ follows the truncated Gaussian distribution $\mathcal{N}_{[0.1,0.9]}(\mu = r_{u,i}^{\text{true}}, \sigma = (1 - r_{u,i}^{\text{true}})/2)$.

• **CRS:** $\hat{r}_{u,i} = 0.2$ if the $r_{u,i}^{\text{true}} \leq 0.6$. Otherwise, $\hat{r}_{u,i} = 0.6$.

**Experimental Details.** For each prediction matrix $\hat{\mathbf{R}} = \{\hat{r}_{u,i}(1) : (u, i) \in \mathcal{D}\}$, the proposed TDR is compared with Naive (Koren et al., 2009), EIB (Hernández-Lobato et al., 2014; Steck, 2010), IPS (Saito et al., 2020; Schnabel et al., 2016), and DR (Wang et al., 2019; Saito, 2020) methods. We obtain the propensities by $1/\hat{p}_{u,i} = (1 - \beta)/p_{u,i} + \beta/p_e$, where $p_e = |\mathcal{D}|^{-1} \sum_{(u,i) \in \mathcal{D}} o_{u,i}$, and $\beta$ is randomly sampled from $[0, 1]$ to introduce noises. Define $\hat{e}_{u,i} = \text{CE}(\sum_{(u,i) \in \mathcal{O}} r_{u,i} w_{u,i}, \hat{r}_{u,i})$, where $w_{u,i} = (1/\hat{p}_{u,i}) / (\sum_{(u,i) \in \mathcal{O}} 1/\hat{p}_{u,i})$, CE denotes the cross entropy loss. For EIB and DR, the imputed error is computed as $\tilde{e}_{u,i} = \hat{e}_{u,i}$, For TDR, $\tilde{e}_{u,i} = \hat{e}_{u,i} + \eta^*(1/\hat{p}_{u,i} - 1)$, where $\eta^* = \arg\min_{\eta} \sum_{(u,i) \in \mathcal{O}} (e_{u,i} - \hat{e}_{u,i} - \eta(1/\hat{p}_{u,i} - 1))^2$. The performance of the estimators is based on the absolute relative error (RE) of the estimated and ideal loss $\text{RE}(\mathcal{L}_{est}) = |\mathcal{L}_{ideal}(\hat{\mathbf{R}}) - \mathcal{L}_{est}(\hat{\mathbf{R}})|/\mathcal{L}_{ideal}(\hat{\mathbf{R}})$, where $\mathcal{L}_{est}$ denotes the estimator to be compared. RE evaluates the accuracy of the estimated loss, and a smaller RE value indicates a higher estimation accuracy.

## 5.2 EXPERIMENT RESULTS (RQ1)

In Table 1, we report the means and standard deviations of the RE of the five estimators for each predicted matrix over 20 times of sampling. On the one hand, the average RE of the IPS, DR and TDR methods is significantly lower than that of the Naive method, verifying the validity of causal-based debiasing methods. The proposed TDR achieves the lowest RE in all settings, attributed to the introduced correction term $\omega_{u,i}$ for estimating $e_{u,i}$, that further reduces the bias of DR. The direct application of the EIB method is even worse than the Naive method, attributed to the challenge to make an accurate estimate of $e_{u,i}$. On the other hand, same as the conclusion of Theorem 1, the standard deviation of the EIB method is significantly lower compared to the IPS and DR methods. The proposed TDR method combines the advantages of the EIB in terms of lower standard deviations than IPS and DR in all settings, reflecting stronger robustness. It can be concluded that the estimation accuracy and robustness of the proposed method are significantly improved compared to the previous methods.

## 6 REAL-WORLD EXPERIMENTS

In this section, we conduct experiments to evaluate the proposed methods on two real-world benchmark datasets containing missing-at-random (MAR) ratings. Throughout, our methods are implemented without uniform data to estimate the propensities, which differs from the existing Naive Bayes approach. We aim to answer the following RQs:

**RQ2.** How do the proposed methods compare with the existing methods in terms of debiasing performance in practice?

**RQ3.** How does the collaborative learning phase design affect the performance of our methods?

**RQ4.** Do our methods stably perform well under different learned propensities?

### 6.1 EXPERIMENTAL SETUP

**Dataset and Preprocessing.** MAR ratings are necessary to evaluate the performance of debiasing methods on real-world datasets. Following previous studies, we take the following two benchmark datasets: **Coat Shopping**[3] has 4,640 MAR and 6,960 MNAR ratings of 290 users to 300 Coats. **Music! R3**[4] has 54,000 MAR and 311,704 MNAR ratings of 15,400 users to 1,000 songs.

**Baselines.** We take the widely used Matrix Factorization (MF) as the base model (Koren et al., 2009), and compare the proposed methods with the following baselines: Base Model (Koren et al., 2009), IPS (Schnabel et al., 2016), SNIPS (Swaminathan & Joachims, 2015), IPS with asymmetric training (IPS-AT) (Saito, 2020), CVIB (Wang et al., 2020b), DIB (Liu et al., 2021), DR (Saito, 2020), DR-JL (Wang et al., 2019), DR-CL, MRDR-JL (Guo et al., 2021), MRDR-CL, where DR-CL and MRDR-CL are performed using the proposed Alg. 1, but without the targeting step update (lines 9-11), also for comparison purpose. In addition, the proposed TDR-based methods include TDR, TDR-JL, and TMRDR-JL implemented by a single targeting step, and TDR-CL and TMRDR-CL implemented by collaborative learning approach as shown in Alg. 1. The real-world experimental protocols and details are provided in Appendix D.

### 6.2 PERFORMANCE COMPARISON (RQ2)

In Table 2, we report the performance of various debiasing methods using MSE, AUC, NDCG@5, and NDCG@10 as evaluation metrics. For previous de-biasing methods, propensity-based IPS, SNIPS, IPS-AT, and information bottleneck-based CVIB and DIB all outperform the base model. The doubly robust methods, such as DR-JL, DR-CL, MRDR-JL, and MRDR-CL, using alternating learning and outperforming DR, which are considered as the most competitive baselines. The proposed TDR estimators are implemented by both single-step and collaborative learning, respectively, based on DR and MRDR as initialized error imputation models, outperforming the baseline methods significantly on all AUC, NDCG@5, and NDCG@10 metrics, attributed to the effectiveness of the introduced nonparametric correction term. It is noted that the collaborative version of TDR achieves the optimal performance both within DR and MRDR, which implements the proposed targeting step repeatedly. The fact that TDR-JL and TMRDR-JL implemented the targeting step only at the final training of the prediction models outperformed DR-JL and MRDR-JL, respectively, further demonstrates the effectiveness of the proposed targeting step to correct imputed errors.

### 6.3 IN-DEPTH ANALYSIS (RQ3, RQ4)

**Ablation Study (RQ3).** To illustrate the specific reasons for the effectiveness of the TDR-CL algorithm, we conduct ablation studies on DR-based and MRDR-based methods, respectively. From Figure 1, DR-CL and DR-JL perform similarly on MSE, AUC and NDCG@5 metrics, and the MRDR approach has similar findings, which indicates that the directly use of collaborative learning approach without targeting steps cannot improve prediction performance. However, for the proposed TDR-CL and TMRDR-CL methods, there is a significant performance improvement compared to

---

[3]https://www.cs.cornell.edu/~schnabts/mnar/
[4]http://webscope.sandbox.Music.com/

Table 2: MSE, AUC, NDCG@5, and NDCG@10 on the MAR test set of Coat and Music. We bold the outperforming DR-based and MRDR-based models. The proposed TDR methods implemented by a single targeting step are marked with $*$ and collaborative learning are marked with $\dagger$.

| | Coat | | | | Music | | | |
|---|---|---|---|---|---|---|---|---|
| | MSE | AUC | N@5 | N@10 | MSE | AUC | N@5 | N@10 |
| Base Model | 0.2448 | 0.7047 | 0.5912 | 0.6667 | 0.2494 | 0.6795 | 0.6353 | 0.7644 |
| + IPS | 0.2389 | 0.7041 | 0.6170 | 0.6852 | 0.2496 | 0.6824 | 0.6409 | 0.7674 |
| + SNIPS | 0.2388 | 0.7061 | 0.6145 | 0.6945 | 0.2493 | 0.6815 | 0.6454 | 0.7701 |
| + IPS-AT | 0.2344 | **0.7320** | 0.6102 | 0.6784 | 0.2480 | 0.6816 | 0.6409 | 0.7667 |
| + CVIB | **0.2201** | 0.7234 | 0.6221 | 0.6991 | 0.2638 | 0.6823 | 0.6483 | 0.7719 |
| + DIB | 0.2334 | 0.7104 | **0.6303** | 0.6986 | 0.2494 | 0.6832 | 0.6348 | 0.7633 |
| + DR | 0.2359 | 0.7031 | 0.6213 | 0.6967 | **0.2420** | 0.6867 | 0.6613 | 0.7791 |
| + DR-JL | 0.2352 | 0.7155 | 0.6183 | 0.6925 | 0.2496 | 0.6853 | 0.6536 | 0.7738 |
| + DR-CL | 0.2358 | 0.7183 | 0.6261 | 0.6927 | 0.2494 | 0.6808 | 0.6334 | 0.7622 |
| **+ TDR**$^{*}$ | 0.2268 | 0.7109 | 0.6300 | **0.7006** | **0.2115** | **0.7044** | **0.7008** | **0.8016** |
| **+ TDR-JL**$^{*}$ | **0.2151** | **0.7236** | **0.6388** | **0.7047** | 0.2577 | **0.7036** | **0.6786** | **0.7884** |
| **+ TDR-CL**$^{\dagger}$ | **0.2119** | **0.7339** | **0.6526** | **0.7112** | 0.2472 | **0.7057** | **0.6758** | **0.7871** |
| + MRDR-JL | 0.2162 | 0.7192 | 0.6360 | 0.7016 | 0.2496 | 0.6842 | 0.6487 | 0.7717 |
| + MRDR-CL | 0.2155 | 0.7200 | 0.6427 | 0.7047 | **0.2494** | 0.6805 | 0.6345 | 0.7623 |
| **+ TMRDR-JL**$^{*}$ | **0.2114** | **0.7278** | **0.6498** | **0.7101** | 0.2557 | **0.7036** | **0.6785** | **0.7884** |
| **+ TMRDR-CL**$^{\dagger}$ | **0.2114** | **0.7316** | **0.6428** | **0.7088** | 0.2473 | **0.7060** | **0.6803** | **0.7902** |

the DR-CL and MRDR-CL methods without targeting steps. This ablation study reveals that the improvement in the proposed TDR-CL and TMRDR-CL originates from the nonparametric correction term of the imputed errors, not from introducing additional model parameters for updating.

**Effect on Learned Propensities (RQ4).** An important fact is that the nonparametric correction term in the TDR estimator is based on given learned propensities. In order to examine whether the proposed targeting steps stably help to improve the prediction accuracy under different learned propensities obtained by setting different clipping threshold, we conducted repeated experiments to quantify the sensitivity of the TDR-CL method to the propensity clipping threshold. From Figure 2, the proposed method outperforms the DR-JL and DR-CL methods in terms of AUC, NDCG@5, and NDCG@10 on all clipping thresholds. The optimal performance is reached when the clipping threshold is equal to 0.15, which is interpreted as achieving the best trade-off between information utilization and robustness.

## 7 RELATED WORK

**Debiasing in Recommendation.** Bias is a common problem inherent in RS (Chen et al., 2020; Wu et al., 2022b), such as popularity bias (Zhang et al., 2021), model selection bias (Yuan et al., 2019), user self-selection bias (Saito, 2020), position bias (Ai et al., 2018), and conformity bias (Liu et al., 2016). Various methods were proposed for unbiased learning. For example, Schnabel et al. (2016) considered the recommendation as treatment and introduced the IPS and self-normalized IPS (SNIPS) methods to debiasing in explicit feedback data. Saito et al. (2020) extended it to the implicit recommendation. Wang et al. (2019) proposed a doubly robust joint learning approach that improved the IPS method. Subsequently, a series of enhanced DR methods were developed, such as MRDR (Guo et al., 2021), Multi-task DR (Zhang et al., 2020), DR-MSE (Dai et al., 2022), BRD-DR (Ding et al., 2022), and SDR (Li et al., 2023c). Li et al. (2023a) proposed a multiple robust method that takes the advantages of multiple propensity and error imputation models. In addition, several new debiasing algorithm are designed via using an extra small uniform dataset (Bonner & Vasile, 2018; Chen et al., 2021; Liu et al., 2020; Wang et al., 2021; Li et al., 2023b). Chen et al. (2020) provided a thorough discussion the recent progress on debiasing tasks in RS. Wu et al. (2022b) established a unified causal analysis framework and gave formal causal definitions of various biases in RS from the perspective of causal inference. Unlike the existing enhanced DR approaches that purse a better bias-variance trade-off, the proposed TDR reduces both the bias and variance and is theoretically guaranteed.

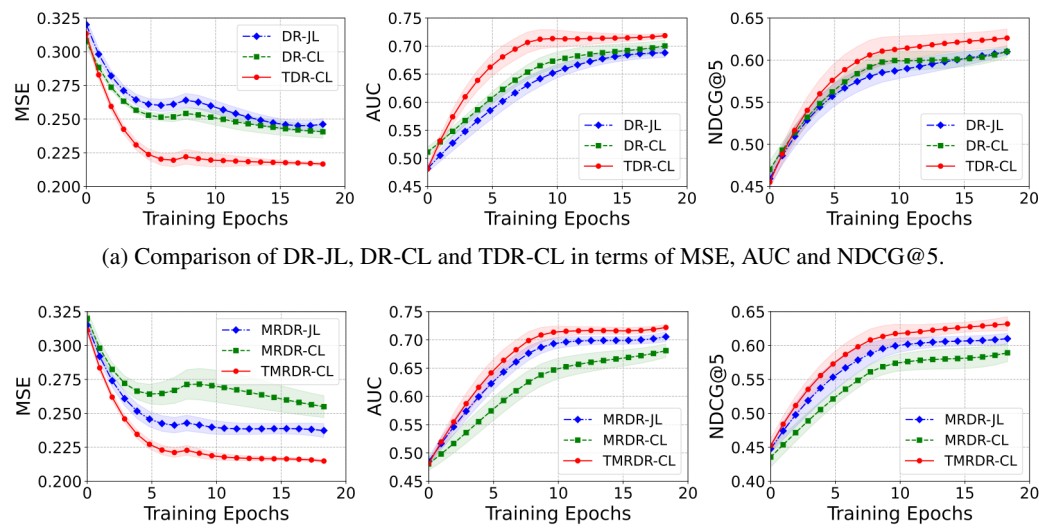

(a) Comparison of DR-JL, DR-CL and TDR-CL in terms of MSE, AUC and NDCG@5.

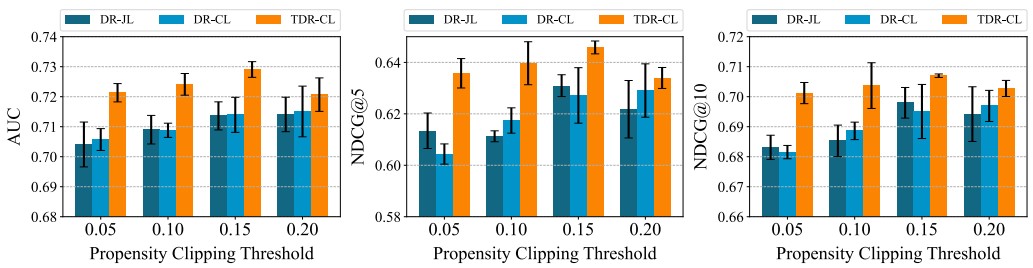

(b) Comparison of MRDR-JL, MRDR-CL and TMRDR-CL in terms of MSE, AUC and NDCG@5.

Figure 1: Ablation studies on DR methods (top) and MRDR methods (bottom), where DR-CL and MRDR-CL skips the targeting steps in TDR-CL and TMRDR-CL.

Figure 2: Learning performance on MAR test set of AUC (left), NDCG@5 (middle), and NDCG@10 (right) with varying levels of propensity clipping threshold.

**Targeted Learning.** Targeted learning is a general framework in causal inference (van der Laan & Rose, 2011) that includes many field-specific approaches to accommodate various scientific problems in different fields, such as survival analysis (Stitelman et al., 2012), genomics (Gruber & van der Laan, 2010), epidemiology (Rose & van der Laan, 2014) and etc. More application scenarios about targeted learning can refer to the two excellent monographs (van der Laan & Rose, 2011; 2018). Shi et al. (2019) proposed adapting neural networks for estimating the average treatment effects based on targeted learning. Different from the existing literature of targeted learning, this paper deals with the estimator and learning problem simultaneously. To the best of our knowledge, this is the first paper that extends targeted learning to the field of debiased recommendation.

## 8 CONCLUSION

In this paper, we propose a TDR estimator for debiased recommendation that enjoys the properties of double robustness, boundedness, low variance, and robustness to small propensities simultaneously. Theoretical analysis shows that TDR can effectively reduce the bias and variance simultaneously for any DR estimator when the error imputation model is less accurate. In addition, we further propose a novel uniform-data-free TDR-based collaborative learning approach that adaptively implements the targeting step, thus making the prediction model more robust. We conducted experiments on both semi-synthetic and real-world data. The superiority of the proposed method is demonstrated when compared with the existing debiasing methods. Throughout, we adopt $1/\hat{p}_{u,i} - 1$ as a key choice of targeting step to satisfy equation (2), which can be regraded as a first-order targeted learning Carone et al. (2014). In future work, we will explore higher-order targeted learning and more effective feature selection in the proposed targeting step.

ETHICS STATEMENT

This work is mostly theoretical and experiments are based on synthetic and public datasets. We claim that this work does not present any foreseeable negative social impact.

REPRODUCIBILITY STATEMENT

Code is provided in Supplementary Materials to reproduce the experimental results.

ACKNOWLEDGMENTS

The work was supported by the National Key R&D Program of China under Grant No. 2019YFB1705601.

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

## A  PROOF OF PROPOSITION 1

Recall that $p_{u,i} = \mathbb{P}(o_{u,i} = 1|x_{u,i}) = \mathbb{E}[o_{u,i}|x_{u,i}]$ and $g_{u,i} = \mathbb{E}[e_{u,i}|x_{u,i}]$, both of them are functions of $x_{u,i}$. Throughout, we maintain the common unconfoundedness assumption (i.e., $r_{u,i}(1) \perp\!\!\!\perp o_{u,i} \mid x_{u,i}$) and the consistency assumption, (i.e., $r_{u,i}(1) = r_{u,i}$ if $o_{u,i} = 1$). All the lower-case letters denote random variables for simplification.

*Proof of Proposition 1.* The property of unbiasedness is obvious. Next, we focus on analysing the variance. Define

$$\sigma^2(x_{u,i}) = \mathrm{Var}(e_{u,i}|x_{u,i}) = \mathbb{E}[(e_{u,i} - g_{u,i})^2 \mid x_{u,i}],$$

then $\mathbb{E}[e_{u,i}^2|x_{u,i}] = \sigma^2(x_{u,i}) + g_{u,i}^2$. The variance of IPS estimator is given by

$$\mathrm{Var}(\mathcal{L}_{IPS}) = |\mathcal{D}|^{-1} \cdot \mathrm{Var}(\frac{o_{u,i}e_{u,i}}{p_{u,i}})$$

$$= |\mathcal{D}|^{-1} \cdot \left[ \mathbb{E}[\frac{o_{u,i}^2 e_{u,i}^2}{p_{u,i}^2}] - \left\{ \mathbb{E}(\frac{o_{u,i}e_{u,i}}{p_{u,i}}) \right\}^2 \right]$$

$$= |\mathcal{D}|^{-1} \cdot \left[ \mathbb{E} \left\{ \frac{\mathbb{E}[o_{u,i}|x_{u,i}] \cdot \mathbb{E}[e_{u,i}^2|x_{u,i}]}{p_{u,i}^2} \right\} - \left\{ \mathbb{E} \left( \frac{\mathbb{E}[o_{u,i}|x_{u,i}] \cdot \mathbb{E}[e_{u,i}|x_{u,i}]}{p_{u,i}} \right) \right\}^2 \right]$$

$$= |\mathcal{D}|^{-1} \cdot \left[ \mathbb{E}[\frac{e_{u,i}^2}{p_{u,i}}] - \left\{ \mathbb{E}(e_{u,i}) \right\}^2 \right]$$

$$= |\mathcal{D}|^{-1} \cdot \left[ \mathbb{E}[\frac{\mathbb{E}(e_{u,i}^2|x_{u,i})}{p_{u,i}}] - \left\{ \mathbb{E}(e_{u,i}) \right\}^2 \right]$$

$$= |\mathcal{D}|^{-1} \cdot \left[ \mathbb{E}[\frac{\sigma^2(x_{u,i}) + g_{u,i}^2}{p_{u,i}}] - \left\{ \mathbb{E}(e_{u,i}) \right\}^2 \right],$$

where the third equation follows by the law of iterated expectations and the unconfoundedness assumption. The variance of DR estimator is derived by

$$|\mathcal{D}| \cdot \mathrm{Var}(\mathcal{L}_{DR}) = \mathrm{Var} \left( g_{u,i} + \frac{o_{u,i}(e_{u,i} - g_{u,i})}{p_{u,i}} \right)$$

$$= \mathrm{Var} \left( e_{u,i} + \frac{o_{u,i} - p_{u,i}}{p_{u,i}}(e_{u,i} - g_{u,i}) \right)$$

$$= \mathrm{Var}(e_{u,i}) + \mathrm{Var} \left( \frac{o_{u,i} - p_{u,i}}{p_{u,i}}(e_{u,i} - g_{u,i}) \right)$$

$$= \mathbb{E}[e_{u,i}^2] - [\mathbb{E}(e_{u,i})]^2 + \mathrm{Var} \left( \frac{o_{u,i} - p_{u,i}}{p_{u,i}}(e_{u,i} - g_{u,i}) \right)$$

$$= \mathbb{E}[\sigma^2(x_{u,i}) + g_{u,i}^2] - [\mathbb{E}(e_{u,i})]^2 + \mathbb{E} \left( \frac{(o_{u,i} - p_{u,i})^2}{p_{u,i}^2}(e_{u,i} - g_{u,i})^2 \right)$$

$$= \mathbb{E}[\sigma^2(x_{u,i}) + g_{u,i}^2] - [\mathbb{E}(e_{u,i})]^2 + \mathbb{E} \left( \frac{\mathbb{E}\{(o_{u,i} - p_{u,i})^2|x_{u,i}\}}{p_{u,i}^2} \cdot \mathbb{E}\{(e_{u,i} - g_{u,i})^2|x_{u,i}\} \right)$$

$$= \mathbb{E}[\sigma^2(x_{u,i}) + g_{u,i}^2] - [\mathbb{E}(e_{u,i})]^2 + \mathbb{E} \left( \frac{p_{u,i}(1 - p_{u,i})\sigma^2(x_{u,i})}{p_{u,i}^2} \right)$$

$$= \mathbb{E}[\frac{\sigma^2(x_{u,i})}{p_{u,i}} + g_{u,i}^2] - [\mathbb{E}(e_{u,i})]^2,$$

where the fifth equation holds by noting that

$$\mathbb{E}\left[ e_{u,i}\frac{(o_{u,i} - p_{u,i})}{p_{u,i}}(e_{u,i} - g_{u,i}) \right] = \mathbb{E}\left[ \frac{\mathbb{E}(o_{u,i} - p_{u,i}|x_{u,i})}{p_{u,i}} \cdot \mathbb{E}\{e_{u,i}(e_{u,i} - g_{u,i})|x_{u,i}\} \right] = 0.$$

Since $\mathbb{E}[o_{u,i}e_{u,i} + (1 - o_{u,i})g_{u,i}] = \mathbb{E}[g_{u,i}] = \mathbb{E}[e_{u,i}]$, we have

$$
\begin{aligned}
|\mathcal{D}| \cdot \mathrm{Var}(\mathcal{L}_{EIB}) &= \mathrm{Var}\left(o_{u,i}e_{u,i} + (1 - o_{u,i})g_{u,i}\right) \\
&= \mathbb{E}\left[\{o_{u,i}e_{u,i} + (1 - o_{u,i})g_{u,i}\}^2\right] - [\mathbb{E}(e_{u,i})]^2 \\
&= \mathbb{E}\left[o_{u,i}e_{u,i}^2 + (1 - o_{u,i})g_{u,i}^2\right] - [\mathbb{E}(e_{u,i})]^2 \\
&= \mathbb{E}\left[p_{u,i}\{\sigma^2(x_{u,i}) + g_{u,i}^2\} + (1 - p_{u,i})g_{u,i}^2\right] - [\mathbb{E}(e_{u,i})]^2 \\
&= \mathbb{E}\left[p_{u,i}\sigma^2(x_{u,i}) + g_{u,i}^2\right] - [\mathbb{E}(e_{u,i})]^2
\end{aligned}
$$

$\square$

## B   PROOF OF THEOREM 1

*Proof of Theorem 1.* The parameter $\eta$ is solved by minimizing

$$
\sum_{(u,i)\in\mathcal{D}} o_{u,i} \cdot \left\{e_{u,i} - \hat{e}_{u,i} - \eta(\frac{1}{\hat{p}_{u,i}} - 1)\right\}^2.
$$

Taking the first derivative of the above loss with respect to $\eta$ and setting it to zero leads to that

$$
\sum_{(u,i)\in\mathcal{D}} o_{u,i} \cdot \left\{e_{u,i} - \hat{e}_{u,i} - \eta(\frac{1}{\hat{p}_{u,i}} - 1)\right\} \cdot (1/\hat{p}_{u,i} - 1) = 0, \tag{4}
$$

which implies that

$$
\sum_{(u,i)\in\mathcal{D}} o_{u,i} \cdot \{e_{u,i} - \tilde{e}_{u,i}\} \cdot (1/\hat{p}_{u,i} - 1) = 0,
$$

namely, the equation (2) holds. This finishes the proof of Theorem 1(a). If $\hat{e}_{u,i}$ already satisfies equation 2), then $\eta = 0$ is a solution of equation (4). Let $\hat{\eta}$ is another solution of equation (4). Since the solution of equation (4) is unique, then $\hat{\eta}$ will converges to 0. This proves the conclusion of Theorem 1(b).

$\square$

## C   PROOF OF THEOREM 2

*Proof of Theorem 2.* The result of Theorem 2(a) is obvious. To show Theorem 2(b). We first claim that if $\hat{e}_{u,i}$ is an accurate estimate of $g_{u,i}$, i.e., $\hat{e}_{u,i} = g_{u,i}$, then it will satisfy equation (2). It holds immediately from the following calculations

$$
\begin{aligned}
&\frac{1}{|\mathcal{D}|} \sum_{(u,i)\in\mathcal{D}} o_{u,i}\{e_{u,i} - \hat{e}_{u,i}\} \cdot (\frac{1}{p_{u,i}} - 1) \\
&= \frac{1}{|\mathcal{D}|} \sum_{(u,i)\in\mathcal{D}} o_{u,i}\{e_{u,i} - g_{u,i}\} \cdot (\frac{1}{p_{u,i}} - 1) \\
&= \mathbb{E}\left[o_{u,i}\{e_{u,i} - g_{u,i}\} \cdot (\frac{1}{p_{u,i}} - 1)\right] \\
&= \mathbb{E}\left[\mathbb{E}(o_{u,i}|x_{u,i}) \cdot \mathbb{E}\{e_{u,i} - g_{u,i}|x_{u,i}\} \cdot (\frac{1}{p_{u,i}} - 1)\right] \\
&= 0.
\end{aligned}
$$

Thus, if $\hat{e}_{u,i}$ not satisfy equation (2), then $\hat{e}_{u,i} \neq g_{u,i}$. Given $\hat{e}_{u,i}$ and $\tilde{e}_{u,i}$, the bias of $\mathcal{L}_{EIB}$ is

$$
\begin{aligned}
\mathrm{Bias}(\mathcal{L}_{EIB}) &= \mathbb{E}[o_{u,i}e_{u,i} + (1 - o_{u,i})\hat{e}_{u,i}] - \mathbb{E}[e_{u,i}] \\
&= \mathbb{E}[(1 - o_{u,i})(\hat{e}_{u,i} - e_{u,i})] \\
&= \mathbb{E}[(1 - p_{u,i})(\hat{e}_{u,i} - g_{u,i})],
\end{aligned}
$$

and the bias of $\mathcal{L}_{TDR}$ is

$$
\begin{aligned}
\text{Bias}(\mathcal{L}_{TDR}) &= \mathbb{E}\left( e_{u,i} + \frac{(o_{u,i} - p_{u,i})}{p_{u,i}}(e_{u,i} - \tilde{e}_{u,i}) \right) - \mathbb{E}[e_{u,i}] \\
&= \mathbb{E}\left( \frac{\mathbb{E}(o_{u,i} - p_{u,i}|x_{u,i})}{p_{u,i}}\mathbb{E}\{e_{u,i} - \tilde{e}_{u,i}|x_{u,i}\} \right) \\
&= \mathbb{E}\left( \frac{0}{p_{u,i}} \cdot (g_{u,i} - \tilde{e}_{u,i}) \right) \\
&= 0.
\end{aligned}
$$

This proves the result of Theorem 2(b).

$\square$

*Biases of DR and TDR.* Given $\hat{p}_{u,i}$ and $\tilde{e}_{u,i}$ for all $(u,i) \in \mathcal{D}$, the bias of TDR is

$$
\begin{aligned}
\text{Bias}(\mathcal{L}_{TDR}) &= \mathbb{E}\left( e_{u,i} + \frac{(o_{u,i} - \hat{p}_{u,i})}{\hat{p}_{u,i}}(e_{u,i} - \tilde{e}_{u,i}) \right) - \mathbb{E}[e_{u,i}] \\
&= \mathbb{E}\left( \frac{\mathbb{E}(o_{u,i} - \hat{p}_{u,i}|x_{u,i})}{\hat{p}_{u,i}}\mathbb{E}\{e_{u,i} - \tilde{e}_{u,i}|x_{u,i}\} \right) \\
&= \mathbb{E}\left( \frac{(p_{u,i} - \hat{p}_{u,i})}{\hat{p}_{u,i}} \cdot (g_{u,i} - \tilde{e}_{u,i}) \right).
\end{aligned}
$$

Similarly, given $\hat{p}_{u,i}$ and $\hat{e}_{u,i}$ for all $(u,i) \in \mathcal{D}$, the bias of DR is

$$
\text{Bias}(\mathcal{L}_{DR}^{(0)}) = \mathbb{E}\left( \frac{(p_{u,i} - \hat{p}_{u,i})}{\hat{p}_{u,i}} \cdot (g_{u,i} - \hat{e}_{u,i}) \right).
$$

$\square$

## D    REAL-WORLD EXPERIMENTAL PROTOCOLS AND DETAILS

**Experimental protocols and details.** For real-world experiments, the following four metrics were considered as the evaluation metrics: *MSE, AUC, NDCG@5,* and *NDCG@10.* For fast convergence in the learning phase, Adam is utilized as the optimizer for all models. We tune the learning rate in $\{0.001, 0.005, 0.01, 0.05, 0.1\}$, weight decay in $[1e-6, 1e-2]$ at 10x multiplicative ratio, and batch size in $\{128, 256, 512, 1024, 2048\}$ for **Coat** and $\{1024, 2048, 4096, 8192, 16384\}$ for **Music! R3**. Specifically for the propensity training, we tune the clipping threshold in $\{0.05, 0.10, 0.15, 0.20\}$. After finding out the best configuration on the validation set, we evaluate the trained models on the MAR test set. Experiments are conducted using NVIDIA GeForce RTX 3090.

