# OpenReview forum: "TDR-CL: Targeted Doubly Robust Collaborative Learning for Debiased Recommendations"
_ICLR.cc/2023/Conference — ICLR 2023 poster_

### Official Review · Reviewer_Z8P2 · 2022-10-24

**Confidence:** 4
**Clarity, Quality, Novelty And Reproducibility:** The paper is clear and is of a good q…
**Correctness:** 3
**Technical Novelty And Significance:** 3
**Empirical Novelty And Significance:** 3
**Recommendation:** 6

**Strength And Weaknesses:**

S1: The idea of applying targeted learning techniques to effectively reduce the bias and variance of existing DR estimators is novel. The paper provides theoretical results showing that the imputed errors obtained with the proposed TDR estimator satisfy two desired properties (validity and preservation) and the proposed TDR estimator has the desired property of being doubly robust.

S2: The paper conducts pretty extensive semi-synthetic experiments and real-world experiments to validate the effectiveness of the proposed TDR estimator and TDR-CL approach, respectively. In the semi-synthetic experiments, the paper simulates a total of six different rating missing scenarios and compares the proposed TDR estimator against four baseline estimators under these diverse scenarios. In the real-world experiments, the paper uses two real-world datasets, compares against a wide range of baseline approaches (IPS, SNIPS, CVIB, DR, DR-JL, etc.), applies the proposed TDR-CL approach to two representative DR approach (DR-JL and MRDR), and computes various evaluation metrics (MSE, AUC, NDCG@5, NDCG@10). The extensive experimental results make the argument that the proposed estimator and approach are more effective than existing approaches much convincing.

W1: In Section PRELIMINARIES, it is not clear why a prediction error is defined as an expectation of the real prediction error given features. It is important to add more explanations and clarifications here.

W2: It would be better if the paper can move the descriptions of the six rating missing scenarios, ONE, THREE, FIVE, Rotate, SKEW, and CRS, from the Appendix to the main text.

**Summary Of The Paper:**

This paper argues that existing doubly robust (DR) estimators usually have a large variance, a large bias, and poor generalization ability given inaccurate error imputation. The paper proposes a novel estimator, called targeted doubly robust (TDR), to simultaneously reduce the bias and variance of existing DR estimators. The paper demonstrates that the imputed errors by the TDR estimator satisfy two important properties called validity and preservation in Theorem 1 and shows that the TDR estimator preserves the desired property of double robustness in Theorem 2. The paper further proposes a semi-parametric collaborative learning approach (TDR-CL) whose key idea is to decompose imputed errors into parametric term and a non-parametric correction term. The paper conducts semi-synthetic experiments to demonstrate that the proposed TDR estimator achieves the lowest absolute relative error among four baseline estimators (Naive, EIB, IPS, and DR) under six rating missing scenarios (ONE, THREE, FIVE, Rotate, SKEW, and CRS). The paper also conducts experiments on two real-world datasets (COAT and YAHOO) and the experimental results show that the proposed approach significantly outperforms baseline approaches under MSE, AUC, NDCG@5, and NDCG@10.



**Summary Of The Review:**

This paper is marginally above the acceptance threshold based on its novelty and significance.

---

> ### Author Response · Authors · 2022-11-16
> **Thanks for your insightful comments and we would like to address your concerns and questions.**
>
> Thank you for the detailed and helpful evaluation of our paper. Please kindly find our responses below.
>
> > _**Q1**: In Section PRELIMINARIES, it is not clear why a prediction error is defined as an expectation of the real prediction error given features. It is important to add more explanations and clarifications here._
>
> **A1**: Thanks for your comments. The $\hat e_{u,i}$ is defined as an estimate of $g_{u,i} = \mathbb{E}(e_{u,i}|x_{u,i})$ instead of $e_{u,i}$,  the reason is that we take $e_{u,i}$ as a random variable. In a regression model, if we fit $y$ using $x$, then for most of the cases (e.g., least square regression), the regression model is essentially estimating the conditional expectation $\mathbb{E}[y|x]$ [1].  In debiased recommendations,
>
>  * For the propensity score model, we use $x_{u,i}$ to predict $o_{u,i}$, thus the propensity score model $\hat p_{u,i}$ is an estimate of $\mathbb{E}[o_{u,i}|x_{u,i}] = P(o_{u,i}=1| x_{u,i}) := p_{u,i}$.
>
>   * For the error imputation model, we use  $x_{u,i}$ to predict $e_{u,i}$, thus the error imputation model $\hat e_{u,i}$ is an estimate of $\mathbb{E}[e_{u,i}|x_{u,i}] := g_{u,i}$.
>
> When studying the theoretical properties of EIB, IPS, and DR, we note that most of literature in debiased recommendation only takes $o_{u,i}$ as the random variable and regards the other variables as fixed constants to simplify the proof [2, 3]. In this paper, the same as [4],  we regard $x_{u,i}$, $o_{u,i}$ and $r_{u,i}(1)$ as random variables, which implies that $e_{u,i}, p_{u,i}$ and $g_{u,i}$ are all random variables.
>
> > _**Q2**:  It would be better if the paper can move the descriptions of the six rating missing scenarios, ONE, THREE, FIVE, Rotate, SKEW, and CRS, from the Appendix to the main text._
>
> **A2**:  Thanks for your advice. Following the previous studies [2, 3, 5], six rating missing scenarios were considered to validate the debiasing performance of the proposed estimator. Due to the page limit, we will move those important experimental settings from the appendix to the main text in our final version.
>
>
> **References**
>
> [1] Hastie, Trevor, Robert Tibshirani, Jerome H. Friedman, and Jerome H. Friedman. The elements of statistical learning: data mining, inference, and prediction. Vol. 2. New York: springer, 2009.
>
> [2] Xiaojie Wang, Rui Zhang, Yu Sun, and Jianzhong Qi. Doubly robust joint learning for recommendation on data missing not at random. In ICML, 2019.
>
> [3] Siyuan Guo, Lixin Zou, Yiding Liu, Wenwen Ye, Suqi Cheng, Shuaiqiang Wang, Hechang  Chen, Dawei Yin, and Yi Chang. Enhanced doubly robust learning for debiasing post-click
> conversion rate estimation. In SIGIR, 2021.
>
> [4] Sihao Ding, Peng Wu, Fuli Feng, Xiangnan He, Yitong Wang, Yong Liao, and Yongdong Zhang. Addressing unmeasured confounder for recommendation with sensitivity analysis. In KDD, 2022.
>
> [5] Schnabel, Tobias, Adith Swaminathan, Ashudeep Singh, Navin Chandak, and Thorsten Joachims. "Recommendations as treatments: Debiasing learning and evaluation." In ICML, 2016.

---

### Official Review · Reviewer_VJ5g · 2022-10-24

**Confidence:** 3
**Correctness:** 4
**Technical Novelty And Significance:** 3
**Empirical Novelty And Significance:** 3
**Recommendation:** 8

**Clarity, Quality, Novelty And Reproducibility:**

The paper is rather clear and well written.

They recall all the desirable properties of existing estimators and compare them in a summary table. This easily highlights what TDR is bringing to the field and its novelty.

In terms of reproducibility, use of public datasets and release of the code is very helpful.


**Strength And Weaknesses:**

Strengths:

The paper is quite clear and enjoyable to read. Motivation of the paper is clearly stated and this is easy to follow for the reader how the authors approached the problem and solved it.

Each of the desirable properties for debiasing estimators (doubly robust, low variance, robust to small properties, without extrapolation and boundedness) is investigated theoretically. Definitions of the concepts are recalled in the appendix.

The theoretical contributions of the paper are significant and are supported by extensive experiments on synthetic and real datasets (best performances are obtained by TDR estimator and its variants).

Weaknesses:

The section on “semi-parametric collaborative learning” might be a little bit dense.

It might be worth also to recall briefly the differences between the competing methods used for comparison in the Section “Real-World Experiments”.

It would be finally helpful for the reader to give a little bit more details on targeted learning and the link with the proposed estimator.

Which stopping criteria and number of steps do you suggest for Algorithm 1 in practice?

Minor (typos):
-p.4: “Specifically, It”

**Summary Of The Paper:**

The authors propose a new estimator for debiasing Recommender Systems called Targeted Doubly Robust (TDR) which addresses some limitations of the Doubly Robust (DR) and Error Imputation Based (EIB) estimators.

They show first that DR has a large variance and is sensitive to small propensities. The variance of DR, while being lower than variance of Inverse Propensity Score (IPS), is higher than variance of Error Imputation Based (EIB) method. The objective is to come up with an estimator with the best of the two worlds: DR and EIB.

By rewriting DR estimator as a sum of the EIB estimator and a correction term, they model the error imputation by any arbitrary function (without any assumptions) so that the correction loss is equal to zero to ensure the lowest variance of EIB but smaller bias than EIB (Theorem 1).

TDR benefits from all other advantages of EIB: boundedness and robustness to small properties. The authors show TDR does not increase the bias compared to DR. Theorem 2 proves its double robustness.

The authors finally show how TDR is used in a collaborative-filtering-based recommender system.


**Summary Of The Review:**

The paper proposes a new estimator TDR for debiasing Recommender Systems which is thoroughly investigated theoretically and well-supported experimentally. TDR obtains best performance on semi-synthetic and real datasets and gathered all the desirable theoretical properties.

---

> ### Author Response · Authors · 2022-11-16
> **Thanks for your insightful comments and we would like to address your concerns and questions.**
>
> Thank you for the detailed and helpful evaluation of our paper. Please kindly find our responses below.
>
> >  _**Q1**: It might be worth also to recall briefly the differences between the competing methods used for comparison in the Section “Real-World Experiments”._
>
> **A1**: Thanks for your helpful suggestions. We have added more comparisons for the previous competing methods in the Section “Real-World Experiments”. Please refer to Section 6.2: -p.4 in our revised version with red texts: "For previous de-biasing methods, propensity-based IPS, SNIPS, IPS-AT, and information bottleneck-based CVIB and DIB all outperform the base model. The doubly robust methods, such as DR-JL, DR-CL, MRDR-JL, and MRDR-CL, using alternating learning and outperforming DR, which are considered as the most competitive baselines."
>
>
> > _**Q2**: It would be finally helpful for the reader to give a little bit more details on targeted learning and the link with the proposed estimator._
>
> **A2**: Thanks for your advice and we apologize for the lack of clarity. We clarify the **relation** and the **difference** between targeted learning and the proposed TDR estimator.
>
> (a) The **relation** between targeted learning and the proposed TDR estimator.
>
> Targeted learning (TL) is a general framework in causal inference that includes many field-specific approaches to answer various scientific problems in different fields. Most of them can be found in these two excellent monographs [1, 2]. Roughly speaking, TL was originally designed to seek an optimal bias-variance trade-off for the initial EIB estimator [1], which can be seen as an improvement of EIB. Thus, it would be a possible solution to reducing the bias/variance of the initial EIB estimator. This article shows that TL can also be used to reduce both the bias and variance of the DR estimator, which is an improvement of DR and is the first DR learning method in recommender systems that can reduce the bias and variance simultaneously. In addition, we also evaluate the proposed method from various important perspectives (e.g., robust to small propensities, boundedness, without extrapolation) that are rarely discussed simultaneously in related works. Furthermore, to the best of our knowledge, this is the first paper that extends TL to the field of debiased recommendation.
>
> (b) The **difference** between targeted learning and the proposed TDR estimator.
>
> Despite we develop the TDR estimator motivated by the earlier work of van der Laan and Rose, the multi-step algorithm in Sec. 4.1 and the collaborative learning in Sec. 4.2 are new. In particular, in our proposed Targeted Doubly Robust Collaborative Learning (TDR-CL), the imputed errors are decomposed into a parametric model estimation and a nonparametric correction term. The parametric and nonparametric parts are updated alternatively to achieve much more accurate prediction error estimation.
>
>
> > _**Q3**:  Which stopping criteria and the number of steps do you suggest for Algorithm 1 in practice?_
>
> **A3**: In our real-world experiments, the stopping criteria and the number of iteration steps of the TDR-CL algorithm are kept the same as in previous studies (e.g., DR-JL [3] and MRDR-DL [4]) for a fair comparison. From Table 3 and Table 4 in Sec. 6, the proposed TDR algorithm has significantly better performance without the cost of additional training time. This is due to the fact that TDR-CL uses both a parametric estimate $\hat e_{u, i}$ and a nonparametric correction term $\omega_{u, i}$ to estimate the prediction error $e_{u, i}$ for a given a prediction model, while the previous methods only use a single parametric model to estimate $e_{u, i}$. This allows our method to estimate $e_{u, i}$ more accurately, which leads to a faster convergence efficiency.
>
>
> > _**Q4**:  Minor (typos): -p.4: “Specifically, It”._
>
> **A4**:  Thanks for pointing this out.  Correction made accordingly.
>
>
> **References**
>
> [1] Mark J. van der Laan and Sherri Rose. Targeted Learning: Causal Inference for Observational and Experimental Data. Springer, 2011.
>
>
> [2] Mark J. van der Laan and Sherri Rose. Targeted Learning in Data Science: Causal Inference for Complex Longitudinal Studies. Springer, 2018.
>
> [3] Xiaojie Wang, Rui Zhang, Yu Sun, and Jianzhong Qi. Doubly robust joint learning for recommendation on data missing not at random. In ICML, 2019.
>
> [4] Siyuan Guo, Lixin Zou, Yiding Liu, Wenwen Ye, Suqi Cheng, Shuaiqiang Wang, Hechang  Chen, Dawei Yin, and Yi Chang. Enhanced doubly robust learning for debiasing post-click
> conversion rate estimation. In SIGIR, 2021.

---

> > ### Comment · Reviewer_VJ5g · 2022-12-02
> > **Thank you very much to the authors for their answers to our reviews and for improving the paper during the rebuttal period**
> >
> > Thank you very much to the authors for their answers to our reviews and for improving the paper during the rebuttal period. The modifications bring valuable content.
> >
> > I read also carefully the other reviews and the corresponding answers.
> > My recommendation is "accept, good paper".

---

### Official Review · Reviewer_k4cv · 2022-10-25

**Confidence:** 3
**Correctness:** 4
**Technical Novelty And Significance:** 3
**Empirical Novelty And Significance:** 3
**Recommendation:** 6

**Clarity, Quality, Novelty And Reproducibility:**

Weaknesses 1, 2 and 3 all include concerns about clarity. In addition, typos exist, e.g., "RQ1. Do the ... estimator ... have ..." should be "RQ1. Does the ... estimator ... have ...".

The quality of the theoretical and experimental work appears to be solid as mentioned above.

The approach does not appear to be particularly novel, but it is an interesting combination of existing ideas.

The code is provided in the supplementary file for reproducibility and sufficient experimental details seem to be provided in the appendices.

**Strength And Weaknesses:**

Strengths:

1. The paper appears to be theoretically strong. Proposition 1, Theorem 1 and Theorem 2 appear to be novel and interesting.

2. Algorithm 1 is itself well-motivated and the emprical results on semi-synthetic data in Table 2 and real data in Tables 3, 4 are quite convincing.

Weaknesses:
1. The authors correctly credit van der Laan and Rose for developing TDR estimators, but they do not relate the multi-step algorithm in Sec. 4.1 on TDR with previous work by van der Laan and Rose. Theorem 1 and Theorem 2 in Sec 4.1 appear to be new, but the reader is not explicitly made aware of this fact.

2. The authors ignore temporal variability in recommender systems and assume that features $x_{ui} $ for a user-item pair are static rather than temporally varying.
Recurrent Recommender Networks, WSDM 2017, shows that this assumption often does not hold for real world recommender systems.

3. The presentation of the material could be improved. For instance, the columns "Without extrapolation" and "Boundedness" in Table 1 are not explained in the paper and one needs to refer to appendices, whereas the symbol \eta in eq (4) changes to \epsilon when we reach Algorithm 1 and its description. The meaning of the columns in Table 2 is not clear until one reads Appendix E.


**Summary Of The Paper:**


The authors propose a novel targeted doubly robust (TDR) algorithm for collaborative learning in recommender systems that alternately optimizes the predictor and propensity estimator encountered in a typical DR estimator along with the error imputation estimate and the step size estimate encountered in a targeted approach. Ablation studies on real data clearly demonstrate the efficacy of the proposed novel components. The authors also convincingly demonstrate improved metrics (AUC, NDCG, etc.) with respect to competing methods.


**Summary Of The Review:**

Given the concerns regarding clarity discussed under weaknesses as well as the second weakness regarding general practical applicability, I weakly recommend acceptance of this paper.

---

> ### Author Response · Authors · 2022-11-16
> **Thanks for your insightful comments and we would like to address your concerns and questions (Part 1 of 2).**
>
> Thank you for the detailed and helpful evaluation of our paper. Please kindly find our responses below.
>
> > _**Q1**: The authors correctly credit van der Laan and Rose for developing TDR estimators, but they do not relate the multi-step algorithm in Sec. 4.1 on TDR with previous work by van der Laan and Rose. Theorem 1 and Theorem 2 in Sec 4.1 appear to be new, but the reader is not explicitly made aware of this fact._
>
> **A1**: We appreciate your recognition of Theorem 1 and Theorem 2 in Sec. 4.1, which are indeed novel, and we apologize for the lack of clarity. We clarify the **relation** and the **difference** between targeted learning and the proposed TDR estimator.
>
> (a) The **relation** between targeted learning and the proposed TDR estimator.
>
> Targeted learning (TL) is a general framework in causal inference that includes many field-specific approaches to answer various scientific problems in different fields. Most of them can be found in these two excellent monographs [1, 2]. Roughly speaking, TL was originally designed to seek an optimal bias-variance trade-off for the initial EIB estimator [1], which can be seen as an improvement of EIB. Thus, it would be a possible solution to reducing the bias/variance of the initial EIB estimator. This article shows that TL can also be used to reduce both the bias and variance of the DR estimator, which is an improvement of DR and is the first DR learning method in recommender systems that can reduce the bias and variance simultaneously. In addition, we also evaluate the proposed method from various important perspectives (e.g., robust to small propensities, boundedness, without extrapolation) that are rarely discussed simultaneously in related works. Furthermore, to the best of our knowledge, this is the first paper that extends TL to the field of debiased recommendation.
>
> (b) The **difference** between targeted learning and the proposed TDR estimator.
>
> Despite we develop the TDR estimator motivated by the earlier work of van der Laan and Rose, the multi-step algorithm in Sec. 4.1 and the collaborative learning in Sec. 4.2 are new. In particular, in our proposed Targeted Doubly Robust Collaborative Learning (TDR-CL), the imputed errors are decomposed into a parametric model estimation and a nonparametric correction term. The parametric and nonparametric parts are updated alternatively to achieve much more accurate prediction error estimation.
>
>
> >  _**Q2**: The authors ignore temporal variability in recommender systems and assume that features_ $x_{u,i}$ _for a user-item pair are static rather than temporally varying. Recurrent Recommender Networks, WSDM 2017, shows that this assumption often does not hold for real-world recommender systems._
>
> **A2**: We thank the reviewer for raising an interesting concern. However, we believe that comparing Recurrent Recommender Networks (RRN, in WSDM 2017) would be outside the scope of our paper, because as stated in RRN paper: "the authors propose Recurrent Recommender Networks (RRN) that are able to predict future behavioral trajectories". Whereas our work is mainly focused on addressing selection bias, i.e., users are free to choose which items to rate, so that the observed ratings are not a representative sample of all ratings.
>
> To evaluate the performance of the prediction models for eliminating selection bias, most of the works [3, 4, 5, 6] use two benchmark datasets, **Coat** and **Yahoo!R3** that both contain missing-not-at-random (MNAR) and missing-at-random (MAR) ratings, as the real-world evaluation criteria. **Notably, Coat and Yahoo do not have timestamps, which further prevents us from studying time-varying selection bias.** In addition, we conducted extensive semi-synthetic experiments using the **ML-100K** and **ML-1M** datasets to illustrate the validity of our proposal. From Table 2 in Sec. 5, the proposed TDR estimator has significantly lower bias and variance compared to DR. For comparison, RRN paper uses datasets containing timestamps (but without MAR ratings), **IMDb** and **Netflix**, and splits them into training and testing sets based on timestamps, which does differ from the purpose of this paper.
>
> Nevertheless, we believe that, as the reviewers point out, it is necessary and important to consider the time-varying selection bias. We leave it to future work, i.e., studying time-varying DR and TDR estimators for eliminating dynamic selection bias.

---

> > ### Author Response · Authors · 2022-11-16
> > **Thanks for your insightful comments and we would like to address your concerns and questions (Part 2 of 2).**
> >
> > > _**Q3**: The presentation of the material could be improved. For instance, the columns "Without extrapolation" and "Boundedness" in Table 1 are not explained in the paper and one needs to refer to appendices, whereas the symbol $\eta$ in eq (4) changes to $\epsilon$ when we reach Algorithm 1 and its description. The meaning of the columns in Table 2 is not clear until one reads Appendix E._
> >
> > **A3**:  Thank you for pointing out this issue. We have unified the notations $\eta$ in eq (4) and $\epsilon$ in Section 4.2 throughout this revision. Due to space constraints in the main text, we will move the detailed explanation and discussion of Table 1 and Table 2 to the main text in our final version.
> >
> >
> > > _**Q4**: Weaknesses 1, 2 and 3 all include concerns about clarity. In addition, typos exist, e.g., "RQ1. Do the ... estimator ... have ..." should be "RQ1. Does the ... estimator ... have ..."._
> >
> >
> > **A4**:  We thank the reviewer for pointing out this issue. We carefully checked and corrected all typos and grammar errors in our revision, and hope this could address your concerns regarding presentation clarity.
> >
> >
> > **References**
> >
> > [1] Mark J. van der Laan and Sherri Rose. Targeted Learning: Causal Inference for Observational and Experimental Data. Springer, 2011.
> >
> > [2] Mark J. van der Laan and Sherri Rose. Targeted Learning in Data Science: Causal Inference for Complex Longitudinal Studies. Springer, 2018.
> >
> > [3] Schnabel, Tobias, Adith Swaminathan, Ashudeep Singh, Navin Chandak, and Thorsten Joachims. "Recommendations as treatments: Debiasing learning and evaluation." In ICML, 2016.
> >
> > [4] Xiaojie Wang, Rui Zhang, Yu Sun, and Jianzhong Qi. Doubly robust joint learning for recommendation on data missing not at random. In ICML, 2019.
> >
> > [5] Siyuan Guo, Lixin Zou, Yiding Liu, Wenwen Ye, Suqi Cheng, Shuaiqiang Wang, Hechang  Chen, Dawei Yin, and Yi Chang. Enhanced doubly robust learning for debiasing post-click
> > conversion rate estimation. In SIGIR, 2021
> >
> > [6] Jiawei Chen, Hande Dong, Yang Qiu, Xiangnan He, Xin Xin, Liang Chen, Guli Lin, and Keping Yang. Autodebias: Learning to debias for recommendation. In SIGIR, 2021.

---

> > > ### Comment · Reviewer_k4cv · 2022-12-01
> > > **Thanks for your response !**
> > >
> > > I wish to thank the authors for their detailed clarifications. Given the complexity of the current approach, it makes sense to relegate any extensions incorporating temporal variability to future work.
> > > My (positive) recommendation remains unchanged.

---

### Decision · Program_Chairs · 2023-01-20

**Decision:**

Accept: poster

**Justification For Why Not Higher Score:**

The work applies existing methods (e.g., targeted learning) to a new context (doubly robust (DR) estimators) for recommender systems. The method is interesting but unclear there is significant novelty for a spotlight or oral.

**Justification For Why Not Lower Score:**

The paper makes a clear contribution in doubly robust (DR) estimators for recommender systems.

**Metareview: Summary, Strengths And Weaknesses:**

The work argues that existing doubly robust (DR) estimators tend to have large variance and bias, with poor generalization ability with inaccurate error imputation. The work then proposes a novel estimator, called targeted doubly robust (TDR), to simultaneously reduce the bias and variance of existing DR estimators. TDR is designed for collaborative learning in recommender systems that alternately optimizes the predictor and propensity estimators. By rewriting DR estimator as a sum of the Error Imputation Based (EIB)  estimator and a correction term, they model the error imputation through an arbitrary function.

The work tackles an important task with an innovative solution. Reviewers agree that the idea of applying targeted learning techniques to effectively reduce the bias and variance of existing DR estimators is novel.

**Note From Pc:**

if the above contains the word "oral" or "spotlight" please see: "oral" presentation means -> notable-top-5% and "spotlight" means -> notable-top-25%. As stated in our emails, we are disassociating presentation type from AC recommendations